# Validation of MSIntuit as an AI-based pre-screening tool for MSI detection from colorectal cancer histology slides

Charlie Saillard [1] ✉, Rémy Dubois[1], Oussama Tchita [1], Nicolas Loiseau[1], Thierry Garcia[2], Aurélie Adriansen[2], Séverine Carpentier[2], Joelle Reyre[2], Diana Enea [3], Katharina von Loga[1], Aurélie Kamoun[1], Stéphane Rossat[2], Corentin Wiscart[1], Meriem Sefta[1], Michaël Auffret[1], Lionel Guillou [1], Arnaud Fouillet[1], Jakob Nikolas Kather [4,5,6] & Magali Svrcek[3,6]

Mismatch Repair Deficiency (dMMR)/Microsatellite Instability (MSI) is a key biomarker in colorectal cancer (CRC). Universal screening of CRC patients for MSI status is now recommended, but contributes to increased workload for pathologists and delayed therapeutic decisions. Deep learning has the potential to ease dMMR/MSI testing and accelerate oncologist decision making in clinical practice, yet no comprehensive validation of a clinically approved tool has been conducted. We developed MSIntuit, a clinically approved artificial intelligence (AI) based pre-screening tool for MSI detection from haematoxylin-eosin (H&E) stained slides. After training on samples from The Cancer Genome Atlas (TCGA), a blind validation is performed on an independent dataset of 600 consecutive CRC patients. Inter-scanner reliability is studied by digitising each slide using two different scanners. MSIntuit yields a sensitivity of 0.96–0.98, a specificity of 0.47-0.46, and an excellent inter-scanner agreement (Cohen's κ: 0.82). By reaching high sensitivity comparable to gold standard methods while ruling out almost half of the non-MSI population, we show that MSIntuit can effectively serve as a pre-screening tool to alleviate MSI testing burden in clinical practice.

Microsatellite Instability (MSI) is a tumour genotype characterised by mismatch errors of repetitive DNA sequences, called microsatellites. It is caused by a deficiency in the DNA mismatch repair (MMR) system, the process whereby errors that occur during DNA replication are fixed. MSI occurs due to MMR malfunction and is therefore a marker of mismatch repair deficiency (dMMR). Found in approximately 15% of the colorectal cancer (CRC) population, MSI plays a crucial role in the clinical management of CRC, with major diagnostic, prognostic, and therapeutic implications. MSI is the hallmark of Lynch Syndrome (LS), the most common form of

hereditary predisposition to develop CRC. MSI tumours are also sensitive to immune checkpoint inhibitor treatments. In 2017, this genomic instability phenotype became the first pan-cancer biomarker approved by the U.S. Food and Drug Administration (FDA), allowing the use of pembrolizumab for patients with MSI unresectable or metastatic solid tumours[1]. Given all the implications of MSI in patient care, many medical organisations such as the National Institute for Health and Care Excellence (NICE) and the National Comprehensive Cancer Network (NCCN), recommend universal screening for MSI status of all newly diagnosed CRC[2,3].

[1]Owkin France, Paris, France. [2]Medipath, Fréjus, France. [3]Department of Pathology, Saint-Antoine Hospital - Sorbonne Université, AP-HP, Paris, France. [4]Else Kroener Fresenius Center for Digital Health, Technical University Dresden, Dresden, Germany. [5]Department of Medicine I, University Hospital Dresden, Dresden, Germany. [6]These authors jointly supervised this work: Jakob Nikolas Kather, Magali Svrcek. ✉e-mail: charlie.saillard@owkin.com

dMMR/MSI can be diagnosed with immunohistochemistry (MMR-IHC) to detect loss of MMR proteins and/or by molecular tests such as polymerase chain reaction (MSI-PCR), or Next Generation Sequencing (NGS). MMR-IHC testing requires excellent tissue fixation, slide preparation time, an experienced pathologist, and consumes tissue material which can be in very limited supply for small tumours. MSI-PCR testing requires specific infrastructure and has generally a longer turnaround time which can delay therapeutic decisions, while NGS remains too expensive to be used routinely. As the number of biomarkers has steadily increased over the last two decades, MMR-IHC and MSI-PCR testing contribute to an ever-increasing workload for pathologists and technicians[4]. Given the global shortage of pathologists worldwide, leveraging AI could ease MSI testing burden by reducing the workload of pathologists[5]. In a 2019 study, we showed that deep learning could accurately detect MSI from H&E slides in CRC[6]. Since then, several studies have presented deep learning-based MSI classifiers from H&E slides in CRC, confirming its potential to complement standard MSI screening methods[7–9].

Despite recent advances, several issues are still preventing AI-based tools for MSI prediction from being used in clinical practice. Most existing studies focus on the area under the ROC curve (AUROC) as their main performance metric. Although useful to compare performances of several machine learning models, this metric can hide a severe lack of generalisation and is not relevant to clinical practice, as pointed out by Kleppe[10]. Here, we refer to model generalisation as the ability of the model to yield consistent sensitivity and specificity in different independent validation cohorts (e.g. with different ethnicities), under different clinical settings (e.g. digitised with different scanners). The AUROC measures the ability of the model to correctly rank patients. In our case, a high AUROC would mean that MSI patients have higher scores (on average) than MSS patients. Therefore, shifting all scores without changing the order would result in the same AUROC. However, in a clinical setting where a threshold is selected and patients are classified as either negative or positive, shifting scores may result in large changes in how patients are classified, altering model's sensitivity and specificity and leading to misdiagnosis. To our knowledge, no studies evaluating performance of AI-based tools to predict MSI from histology slides have solved the issue of model generalisability in such a way as to enable its use in clinical routine. In this study, we propose to focus on sensitivity, specificity and negative predictive value to evaluate diagnostic accuracy of MSIntuit™ CRC (MSIntuit), an AI pre-screening solution that enables an early rule-out of non-MSI patients using H&E slides from primary resected colorectal tumour. MSIntuit outputs either "MSS-AI" (no further testing needed) or "Undetermined" (standard MSI test required) (Fig. 1a). Importantly, an MSI pre-screening tool used as a rule-out test must have a very high sensitivity. We therefore propose a method that guarantees the sensitivity is maintained at new pathology laboratories.

Self-supervised learning (SSL) has emerged in the computer vision field as a powerful method to learn rich vector representations from images. SSL consists of training a feature extractor to solve a "pretext task", that is a task that does not require human annotations, as

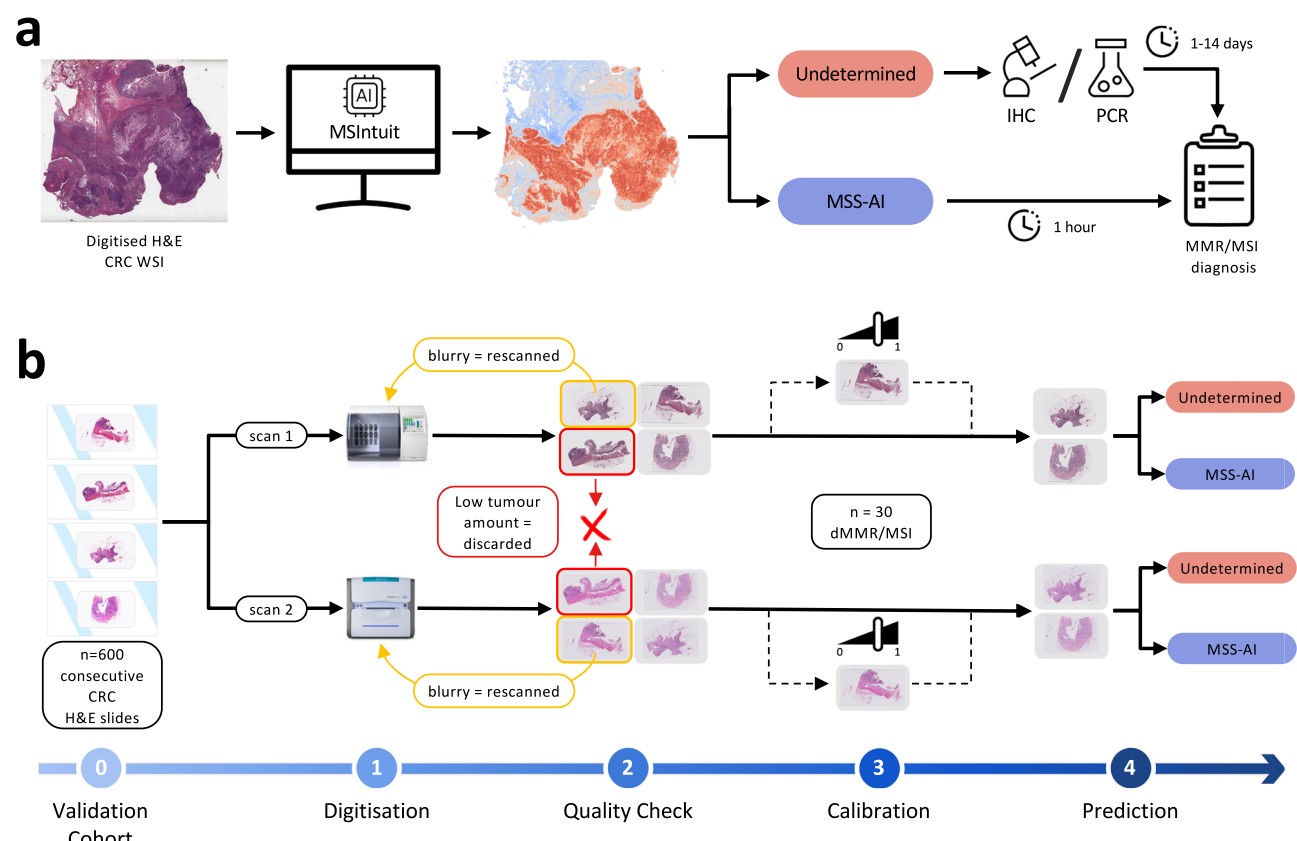

**Fig. 1 | Clinical workflow and blind validation methodology. a** Clinical workflow of MSI screening with MSIntuit. Using a routine H&E slide of CRC, MSIntuit outputs if the patient is likely to be MSI (Undetermined) and should receive a confirmatory test (MMR-IHC and/or MSI-PCR), or not (MSS-AI). By ruling out a significant fraction of non-MSI patients, the workload of pathologists is reduced and the MSI screening is accelerated. **b** H&E slides of 600 consecutive resected CRC specimens were collected and digitised on two scanners, Phillips UFS and Ventana DP200, resulting in two sets of slides: MPATH-UFS and MPATH-DP200 (step 1). For each cohort, the same pipeline was then applied: an automated quality check discarded slides that did not match criteria (large blurry regions, too few tumour). Slides with large blurry regions were rescanned (step 2). Next, 30 dMMR/MSI WSIs were selected randomly and used to define an appropriate threshold (step 3). Finally, MSIntuit prediction was performed on the remaining slides using the threshold defined in the aforementioned step to classify patients into two categories: MSS-AI and Undetermined (step 4).

opposed to traditional supervised learning. Such tasks can be reconstructing a part of the image which is masked, or producing similar representations for two augmented versions of the same image[11,12]. MSIntuit leverages a feature extractor tailored for histology, trained on four million colorectal cancer pathology images with SSL.

In this work, we perform a blind clinical validation of MSIntuit on a large external cohort of 600 consecutive resected CRC cases. We find that using MSIntuit as a pre-screening tool can rule out almost half of the non-MSI population, thus easing MSI screening. Our tool includes an automatic slide quality check and addresses the issue of defining an operating threshold with a calibration step, making it directly applicable to clinical practice. We also address key questions for use in clinical routine by studying MSIntuit's intra and inter scanner variability, as well as inter-block variability.

## Results

### Quality check and calibration as preliminary steps for a clinical-ready AI-based tool

An automated quality check (QC) was first performed on MPATH-DP200 and MPATH-UFS cohorts to set aside slides that did not meet the tool requirements. This step allows us to automatically detect slides that were not properly scanned and contained large blurry regions, which could impact the final prediction score. Interestingly, these blurry slides were not noticed by the pathologists because it was only visible at a high magnification level (Supplementary Fig. 1). The QC was able to identify these slides quickly, without the need for manual examination. As a result, 3% of MPATH-DP200 slides and 2% of MPATH-UFS slides were rescanned. The second step of QC allowed to detect slides with too little tumour tissue (<6 mm$^2$): 5% and 2% of the slides were discarded on MPATH-DP200 and MPATH-UFS cohorts, respectively. As a result of this preprocessing, we obtained $n = 537$ (MSI: 83) and $n = 554$ (MSI: 86) slides for MPATH-DP200 and MPATH-UFS cohorts respectively.

Any deep learning system in pathology requires a threshold to convert continuous prediction values into actionable categories. To address the issue of variations in data acquisition protocols that may impact deep learning model prediction distributions, we used a calibration strategy to ensure a sensitivity between 0.93 and 0.97 was obtained (see methods section "Calibration step", Fig. 1b). This step enabled MSIntuit to reach high sensitivity, which is critical in a clinical setting, without sacrificing MSIntuit' specificity, which is important to guarantee the tool's clinical utility. This process led us to choose a threshold of 0.20045 on the MPATH-UFS dataset and 0.20202 on the MPATH-DP200 dataset. The similarity of the two thresholds suggests that the variations between UFS and DP200 scanners did not meaningfully impact MSIntuit predictions, despite the model having been trained on data collected using another scanner (Leica Aperio).

### MSIntuit performance was boosted using self-supervised learning, allowing it to rule out almost half of the non-MSI population with high sensitivity

During model development, we found that using a feature extractor pre-trained with SSL on millions of histology tiles yielded a performance improvement. To illustrate this, we compared this approach against two other feature extraction approaches, keeping the other steps of MSIntuit pipeline unchanged (matter detection, QC, ..., see Methods section). The first approach consisted of using an extractor pre-trained on ImageNet dataset, while the second consisted of using an extractor pre-trained on 100,000 colorectal cancer images to identify nine tissue classes[13]. Although ImageNet only contains natural images, the first method has been used widely in the computational pathology community because there is no dataset of histology images equivalent to ImageNet in terms of size (1.2 million images) and annotation diversity (1000 classes). The second approach has the advantage of leveraging a feature extractor directly trained on

colorectal cancer images. However, this feature extractor has seen a lower number of distinct images during training, which may impact its representation capacity. We also compared MSIntuit against iDaRS, a recently published method which finetunes an extractor pre-trained on ImageNet to predict MSI using an innovative weakly supervised approach[8]. MSIntuit's approach largely outperformed the other methods by more than eight AUROC points on PAIP, MPATH-DP200 and MPATH-UFS (Supplementary Tables 1, 2). Including frozen slides in the training set and applying our model to the whole-slide (not just the tumour content) also yielded small performance improvements (Supplementary Tables 3, 4).

Following QC and calibration, predictions of MSI status were generated from the histology slides and resulted in a sensitivity of 0.98 (95% CI: 0.95–1.0), an NPV of 0.99 (0.98–1.0), and a specificity of 0.46 (0.42–0.50) on the MPATH-DP200 cohort, and a sensitivity of 0.96 (0.91–0.98), an NPV of 0.98 (0.97–0.99), and a specificity of 0.47 (0.43–0.51) on the MPATH-UFS cohort (Tables 1, 2). On both cohorts, MSIntuit was therefore able to correctly identify the majority of MSI patients while ruling out almost half of the non-MSI population and enriching the remaining population to screen in MSI patients by 60%. This shows the robustness of our calibration approach and that our model generalises well on an independent cohort and across two different scanners not used during training.

We assessed the ability of MSIntuit to detect unusual isolated losses of PMS2 and MSH6 mutations, which were found to cause discordance between MMR-IHC and PCR-MSI[14]. MSIntuit reached a sensitivity of 0.91 (respectively 0.91) and 0.67 (respectively 0.72) on MPATH-DP200 (respectively MPATH-UFS) to detect PMS2 and MSH6 losses, respectively (Supplementary Table 5). Because less than ten cases displayed these mutation patterns, further evaluation with larger sample sizes should be carried out to confirm these numbers.

To assess the importance of the QC, we looked at the performance of MSIntuit on MPATH-DP200 after removing this step. Without discarding the slides with too few tumour, performance decreased to an AUROC of 0.86, a sensitivity of 0.96, a specificity of 0.45 and a NPV of 0.98 (Supplementary Table 6). No significant difference in performance was observed when the slides with large blurry areas were not digitised again, nevertheless, we observed small differences in score. Median prediction for blurry (respectively rescanned) slides was of 0.29 (respectively 0.21) for MSS cases and 0.55 (respectively 0.56) for MSI cases (Supplementary Fig. 2).

### MSIntuit reached excellent agreement on two scanners, and is repeatable across multiple rescanning of the same slide

Several studies have shown that different scanners induce variations on the digital images generated, which can hamper the development of computational pathology (CP) tools[15,16]. Given that various scanner models are used across medical centres, it is crucial that CP tools can handle these data acquisition variabilities. Results presented in the previous section show that MSIntuit generalises well to scanners not

**Table 1 | Confusion matrix of MSIntuit on MPATH-DP200 cohort**

|         | MSS-AI | Undetermined |
|---------|--------|--------------|
| non-MSI | 208    | 246          |
| MSI     | 2      | 81           |

**Table 2 | Confusion matrix of MSIntuit on MPATH-UFS cohort**

|         | MSS-AI | Undetermined |
|---------|--------|--------------|
| non-MSI | 218    | 250          |
| MSI     | 4      | 82           |

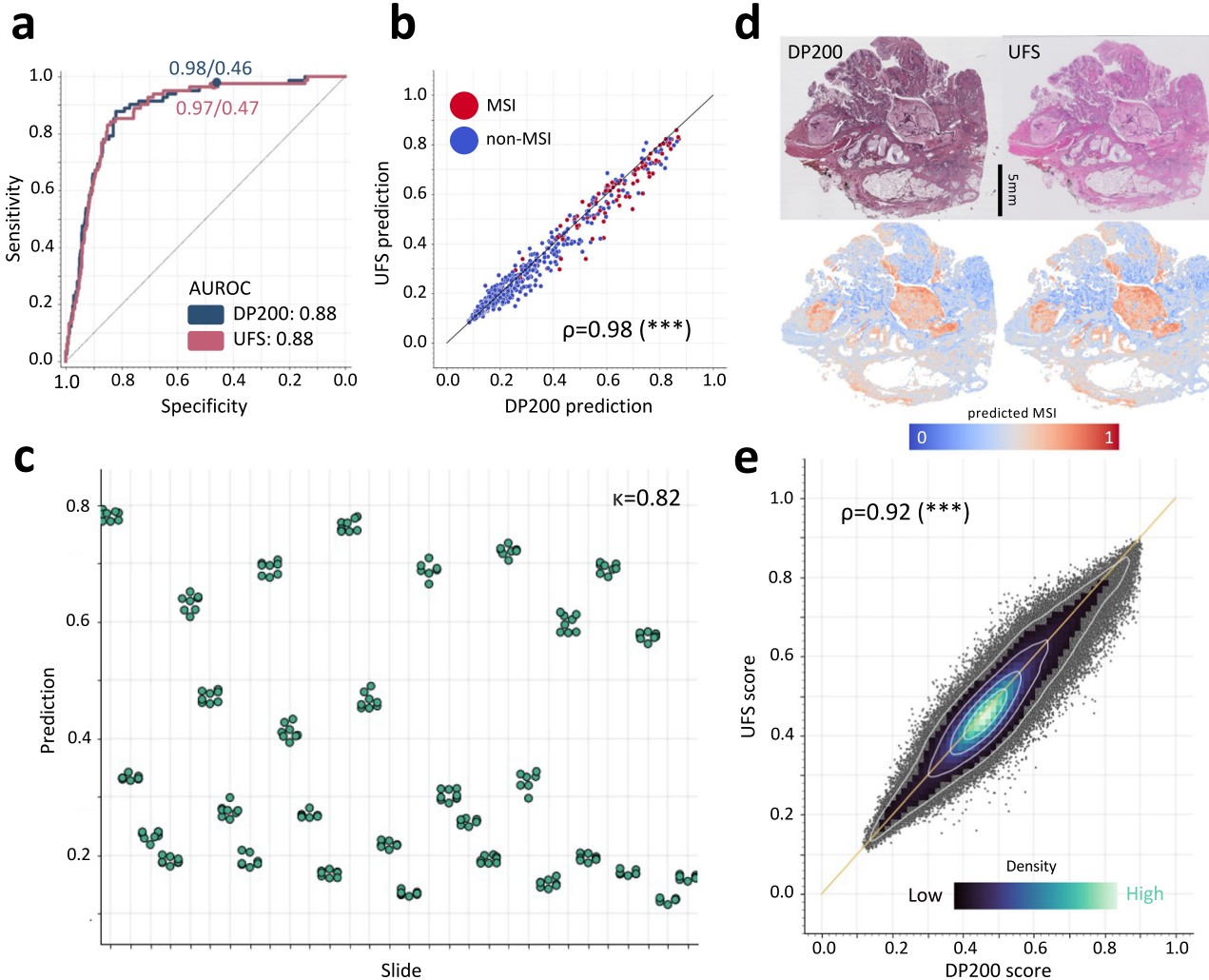

**Fig. 2 | Robustness to scanner variations. a** ROC curves of MSIntuit performance on MPATH-DP200 and MPATH-UFS cohorts. To compare performance on the exact same set of patients, we kept the subset of patients that passed QC on the two sets of slides ($n = 536$), and obtained an AUROC of 0.88 on both scanner, **b** Correlation of the predictions on the same slides on the UFS/DP200 scanners resulting in Pearson's correlation of 0.98 (two-sided $t$ test $p < 0.001$), **c** Prediction distribution for 30 slides, where each slide was digitised 8 times with the UFS scanner. Fleiss' Kappa of 0.82 was obtained, showing an almost perfect agreement of the tool between the different digitisation of the same slide. **d** Heatmaps showing MSI score

for each 112 × 112 µm tile for one representative slide digitised with two scanners, **e** Correlation of tile MSI scores on DP200 and UFS scanner. MSIntuit outputs a score for each tile, hence we also analysed the concordance of tile scores for a subset of 20 slides digitised with the two scanners ($n = 272{,}527$ tiles). A Pearson's correlation of 0.92 was obtained (two-sided $t$ test $p < 0.001$). The colormap representing the spatial density of points indicates that most tile scores were close to the diagonal, showing that tile scores were highly concordant. Source data are provided as a Source Data file.

used during model training. To further study this potential issue, we assessed the impact of digitisation variations on MSIntuit by comparing the results obtained on MPATH-DP200 and MPATH-UFS cohorts, which were composed of the exact same slides digitised with these two different scanners. We first compared the results obtained on the exact same set of slides across the two scanners, and found that model performances were very close with an identical AUROC of 0.88 (95% CI: 0.85–0.91) (Fig. 2a). Additionally, correlation of predictions across the two scanners was very strong with a Pearson's R of 0.98 ($p < 0.001$, Fig. 2b), with an overall mean inter-scanner score difference of 0.01 (95% CI: −0.06–0.09) (Supplementary Fig. 3). Interestingly, the correlation was substantially lower using the machine learning approaches mentioned in the previous section (ImageNet: $R = 0.82$, NCT-CRC-100K: $R = 0.70$, iDaRS: $R = 0.58$) (Supplementary Fig. 4). As MSIntuit feature extractor was trained specifically to produce similar representations under heavy data augmentations, we believe that this could explain the enhanced robustness of MSIntuit to scanner variations. We measured the agreement of the categories output by MSIntuit on the

two scanners: an almost perfect agreement was observed with a Cohen's Kappa of 0.82. As MSIntuit also outputs one score per tile (representing the likelihood of the tile belonging to a MSI slide), we further assessed the model's robustness to the scanner at this finer level (Fig. 2d). 272,527 tiles of 20 slides sampled randomly (MSI: 10, non-MSI: 10) were used and a score was generated for each of them on the two scanners. A very strong correlation was observed with a Pearson's R of 0.92 ($p < 0.001$, Fig. 2e). Finally, we assessed the intra-scanner reliability of our tool by looking at the process of digitisation: 30 slides were digitised 8 times on the UFS scanner. Agreement of the tool across the different digitisations was almost perfect with a Fleiss' Kappa of 0.82 (Fig. 2c).

**MSIntuit results were consistent across slides obtained from different regions of the tumour**

Since several slides are usually available for each patient that may highlight different aspects of the tumour, some criteria are needed to ensure that the slide processed by the tool is representative of the

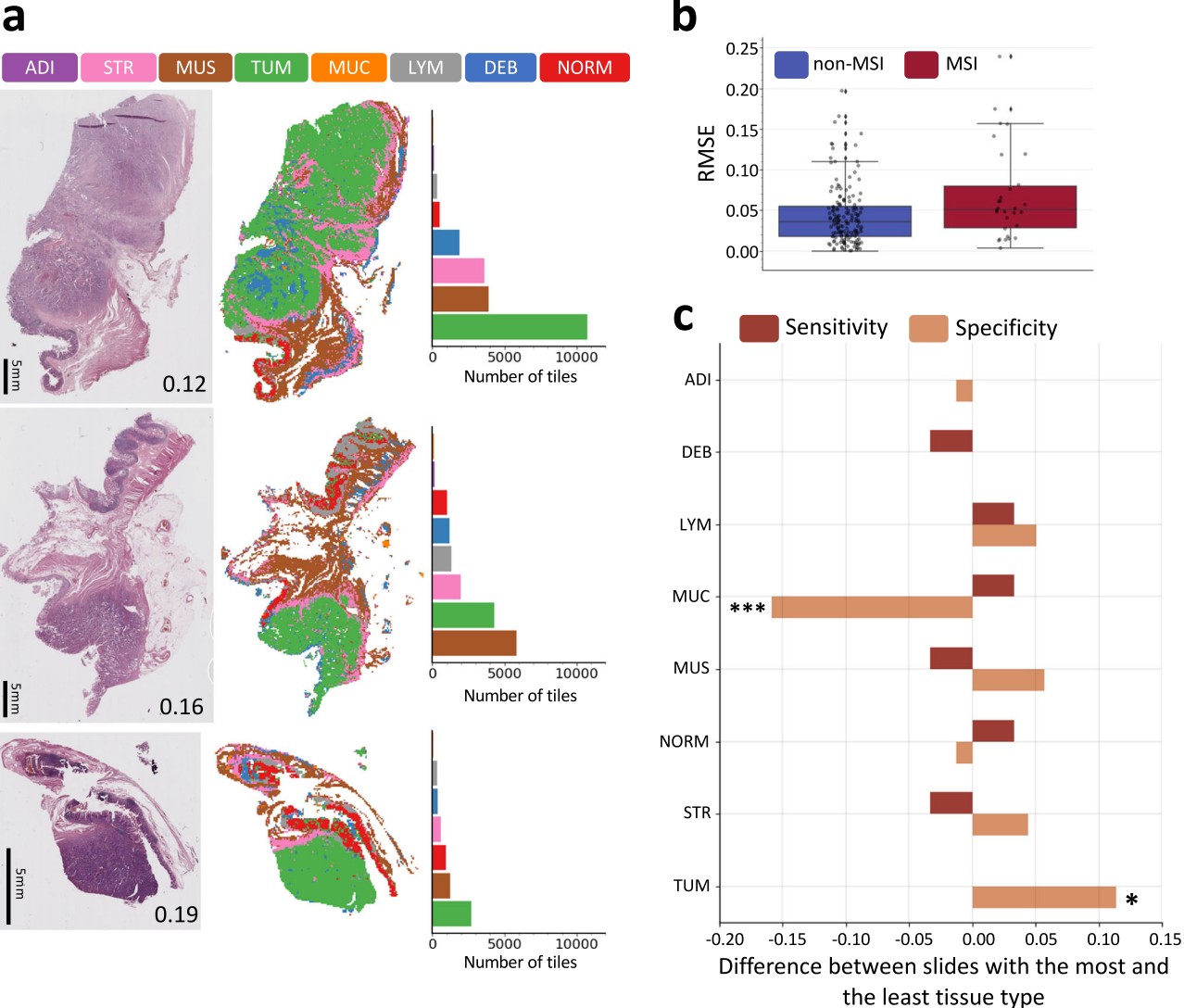

**Fig. 3 | Impact of slide selection on MSIntuit. a** Impact of tumour heterogeneity on MSIntuit prediction on a representative non-MSI case. Left: 3 slides picked from different blocks of the same tumour. The number on the bottom right corner of each slide corresponds to the tool's prediction for the given slide. Middle: segmentation maps using a model trained to categorise tissue into one of the 8 following categories: adipose (ADI), debris (DEB), lymphocytes (LYM), mucin (MUC), smooth muscle (MUS), normal colon mucosa (NORM), cancer-associated stroma (STR), colorectal adenocarcinoma epithelium (TUM). Right: number of tiles belonging to each category. The slide with the largest amount of tumour was the closest to 0; as this patient is non-MSI, this slide gave the best prediction. **b** MSIntuit's predictions variability due to using different slides available for the same patient, for 200 patients with one to four additional slides of the tumour available. Root mean squared errors (RMSE) of slide prediction and the average of

the corresponding patient' slides were computed and resulted in an average RMSE of 0.04 and 0.07 for non-MSI and MSI patients respectively. Center corresponds to the median, lower, and upper hingers to the first and third quartiles, whiskers to the hist/lowest value no further than 1.5 × IQR (inter-quartile range). **c** Difference of sensitivity/specificity when selecting slide with the highest and lowest amount of each tissue type was computed for the 200 tumours with multiple slides. Choosing the slide with the lowest amount of mucin and largest amount of tumour resulted in a significantly better specificity (+15 points, $p < 0.001$ and +10 points respectively, $p < 0.05$). Other categories were not significantly associated with a better sensitivity or specificity. P-values were computed using a McNemar test of homogeneity. No adjustment for multiple comparisons were made. Source data are provided as a Source Data file.

tumour. Guidelines are detailed in the Supplementary methods. We showed in the previous section that good performance was obtained with these guidelines. We further explored the consistency of our tool with respect to the region of the tumour processed by digitising additional slides from one to four other blocks for a subset of 200 out of the 600 tumours of MPATH-DP200 cohort. Average difference of predictions for different slides of the same tumour was low for both non-MSI and MSI patients with a root mean square error of 0.04 and 0.07, respectively (Fig. 3b), indicating that the MSIntuit prediction score is consistent between tumour blocks. For the same set of 200 tumours, we also assessed which slide of the tumour should be selected to maximise MSIntuit performance and found that selecting

the slide with the lowest amount of mucin and largest amount of tumour resulted in a significantly better specificity (+15 points, $p < 0.001$ and +10 points respectively, $p < 0.05$, Fig. 3c).

## MSIntuit provides interpretable results for pathologists

MSIntuit outputs a score for each tile, enabling to retrieve the regions of interest found by the algorithm. Heatmaps and most predictive regions of two representative slides of MPATH-DP200 are provided in Fig. 4. Five pathologists (T.G., A.A., S.C., J.R., D.E.) reviewed the 400 tiles most predictive of MSI ($n = 200$) and non-MSI ($n = 200$) statuses, blinded to their scores (more information in Supplementary methods). We found that the majority of tiles predictive of both MSI and non-MSI

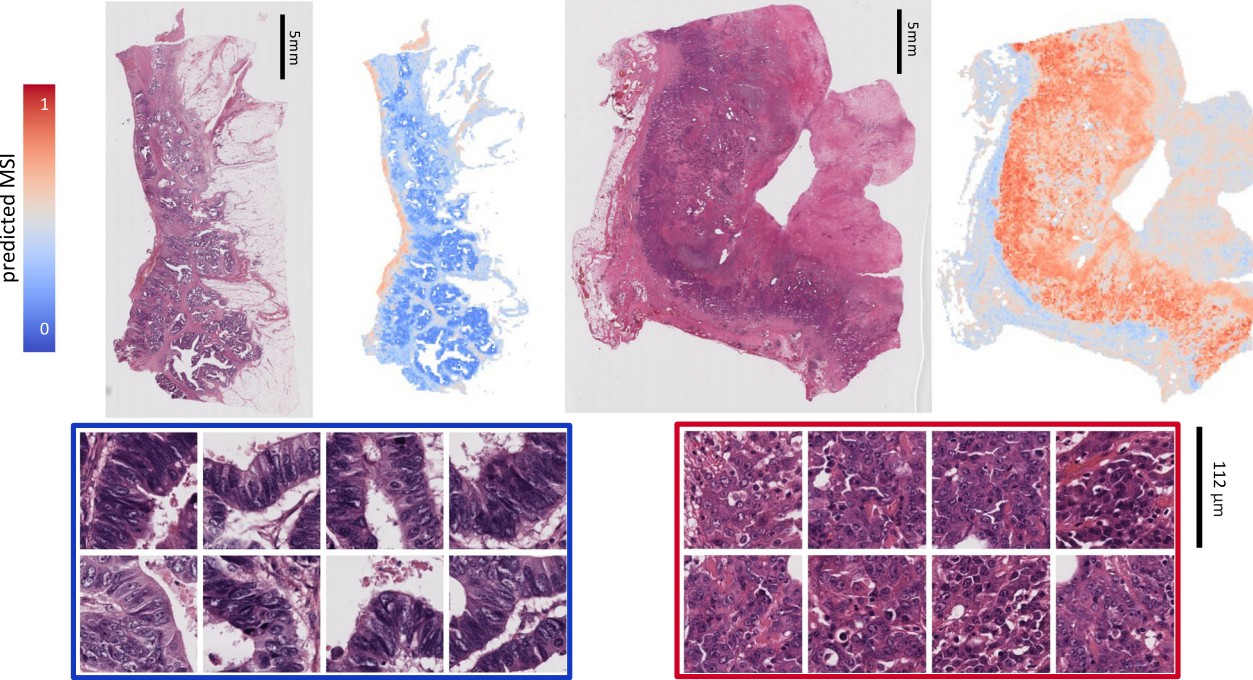

**Fig. 4 | Top: MSIntuit prediction heatmaps showing MSI score for each 112 × 112 μm tile on representative non-MSI (left) and MSI (right) cases.** Bottom: Corresponding most predictive regions of non-MSI (left) and MSI regions (right). Regions predictive of MSI displayed poor differentiation, tumour infiltrating lymphocytes while regions predictive of non-MSI were well differentiated tumour glands.

contained tumour cells, with MSI: 70%, non-MSI: 60%. Tiles predictive of MSI were associated with inflammation (MSI: 50%, non-MSI: 13%, $p < 0.001$) and mucin (MSI: 28%, non-MSI: 6%, $p < 0.001$). Tiles predictive of non-MSI were associated with normal glands (MSI: 4%, non-MSI: 26%, $p < 0.001$) (Fig. 5a). These observations are in line with the histological patterns previously described as associated with MSI tumours, as well as the interpretability analyses of deep learning models predicting MSI[7,8,17,18]. Interestingly, although the presence of mucin was predictive of MSI according to MSIntuit, we showed in the previous section that it could also cause false positive results. This is also in line with the findings from multiple studies which reported that mucinous tumours were represented in both non-MSI and MSI tumours but were over-represented in the latter subgroup[17,19].

Regions predictive of MSI also included inflammation outside the tumour area (25%), which may explain why better performance was obtained considering the whole-slide and not just the tumour content.

A pathologist (D.E.) thoroughly reviewed the slides of the four MSI patients missed by our tool. Interestingly, these cases had a well differentiated glandular architecture and did not display any of the patterns known to be associated with MSI (Fig. 5b).

To better quantify the information brought by our tool against patterns known to be associated with MSI, we further compared the performance of MSIntuit against MSPath, a scoring system including clinical and pathological variables (age at diagnosis, anatomical site, histologic type, grade, presence of Crohn-like reaction, presence of tumour infiltrating lymphocytes)[20]. On a subset of 202 cases from MPATH-DP200 cohort, MSIntuit outperformed MSPath with an AUROC of 0.88 (MSPath: 0.75, Fig. 5c). The two algorithms both reached a sensitivity of 0.97, but MSIntuit reached a better specificity of 0.45 (MSPath: 0.40). More information about the comparison assessment and features distribution can be found in the Methods section *"Comparison of MSIntuit with MSPath scoring system"* and Supplementary Table 7. Interestingly, both MSPath and MSIntuit were found to be statistically significant predictors of the MSI status in multivariate analysis (Supplementary Table 8). A simple dichotomic classifier combining both scores yielded a sensitivity of 0.95 and a specificity of 0.67 (Supplementary Table 9). This shows that MSIntuit brings additional information to a scoring system using clinical and pathological features known to be associated with MSI.

## Discussion

In this study, we reported the development and blind validation of MSIntuit, an AI-based tool that can be used in clinical practice for MSI pre-screening from routine H&E slides of CRC patients. Used as a pre-screening tool, MSIntuit can rule out almost half of the non-MSI population while correctly classifying more than 96% of dMMR/MSI patients, on par with current gold standard methods (92–95%).

The major strength of the study is the blind validation of the model on 600 consecutive CRC cases diagnosed across nine different pathology labs in the span of two years, thus reducing the risk of selection bias. Most importantly, prediction and performance assessment procedures were pre-specified and the validation was performed in a one-shot fashion to avoid the risk of overfitting. Finally, MSI-PCR was used to confirm doubtful cases of MMR-IHC to ensure the accuracy of the dMMR/MSI labels and validation was done on two different scanners not used during model training. Altogether, we believe this demonstrates the strength of our validation, as well as the robustness of MSIntuit.

A key technical strength of the approach is the use of SSL to extract features from the histology images. Using this method, we were able to train a feature extractor tailored for histology on four million CRC histology images without the need for any labels. As already pointed out by previous studies, we observed that such methods were more robust to scanner variations and largely outperformed feature extractors pretrained on ImageNet dataset for MSI prediction task, an approach still widely used in medical imaging[21–23].

We also showed that MSIntuit outperformed MSPath, a scoring system which uses clinical and pathological variables known to be associated with MSI. It is worthy to note that this system is subject to interobserver variability and requires a time-consuming assessment of several histological features by pathologists, which explains why it is not used in clinical practice given their high workload. Importantly, the

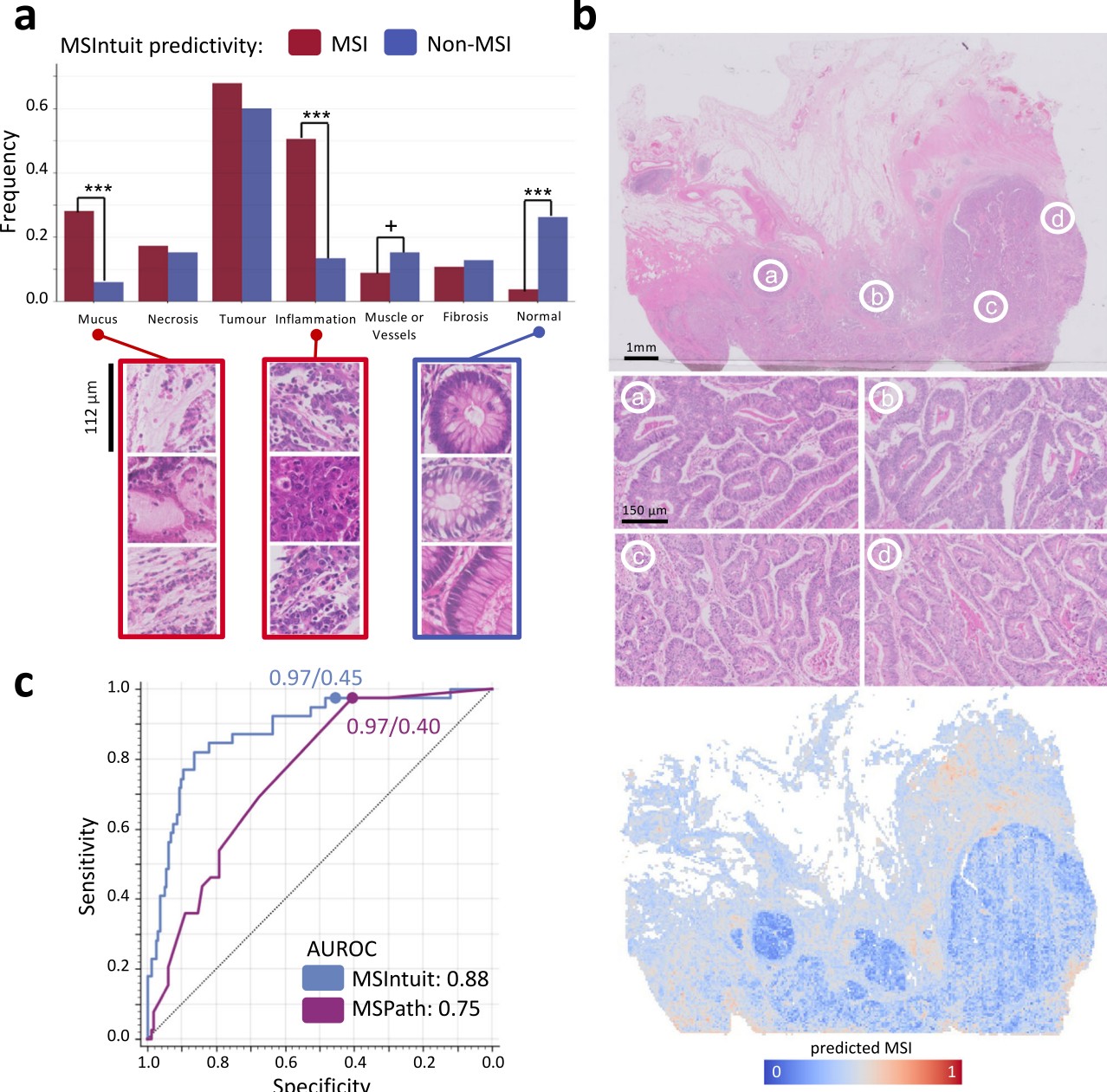

**Fig. 5 | Model interpretability and comparison with a clinico-pathological MSI scoring system. a** Proportion of histology patterns associated with non-MSI and MSI according to MSIntuit. Four pathologists reviewed the 400 tiles most predictive of MSI ($n = 200$) and non-MSI ($n = 200$) statuses, blinded to their scores. Majority of tiles predictive of both MSI and non-MSI contained tumour cells, with MSI: 70%, non-MSI: 60%. Tiles predictive of MSI were associated with inflammation (MSI: 50%, non-MSI: 13%, $p < 0.001$) and mucin (MSI: 28%, non-MSI: 6%, $p < 0.001$). Tiles predictive of non-MSI were associated with normal glands (MSI: 4%, non-MSI: 26%, $p < 0.001$). *P*-values were computed using a two-sided test for proportions based on normal (z) test. No adjustment for multiple comparisons were made. **b** Top: Example slide of an MSI patient which was incorrectly classified by MSIntuit (score: 0.13, bottom 10%). Middle: The slide displayed well differentiated regions, without any histological patterns known to be associated with MSI. Bottom: corresponding MSIntuit heatmap. **c** Performance comparison of MSIntuit and MSPath on a subset of 202 patients from MPATH-DP200 cohort. MSIntuit (respectively MSPath) reached an AUC, Sensitivity and Specificity of 0.88, 0.97, 0.45 (respectively 0.75, 0.97, 0.40). Source data are provided as a Source Data file.

combination of MSIntuit and this system yielded a performance improvement, suggesting that MSIntuit brings additional information to variables known to be associated with MSI.

With recent In-vitro Diagnostic 'Conformité Européenne' (CE-IVD) certification, this study paves the way for MSIntuit use in clinical routine. A key objective of this study was to ensure that the MSIntuit tool could be deployed in clinical centres. To examine the impact of using different scanners at different sites, we digitised 600 slides with two different scanners. We found that MSIntuit was robust to these variations and reached almost perfect agreement and similar performances on both DP200 and UFS scanners. To ensure that the sensitivity of the MSIntuit tool would be maintained across each new clinical centre, we developed a calibration approach outperforming existing methods for MSI prediction[18]. We found that setting a clinically relevant operating threshold could be done by using 30 MSI slides.

For pathology labs pressured to support an ever-increasing number of biomarkers to screen for while facing a growing shortage of pathologists, the advent of AI-enabled solutions that ease and

optimise biomarker screening seems necessary. Using MSIntuit in clinical routine, pathologists could rapidly rule out almost 50% of non-MSI cases prior to any standard MSI testing technique. With MMR-IHC turnaround time varying between two to seven days in different clinical settings and MSI-PCR results delivery that can take more than a week, pre-screening for non-MSI patients with an AI-enabled solution in a few hours holds a real potential of time-savings, both for pathologists and patients. This approach will have a direct impact on oncologist decision making and help bring the best treatment to patients sooner. It could also optimise costs and organisation of MSI testing in pathology labs, especially for countries applying universal MSI screening. As highlighted by Kacew et al., a medico-economic evaluation of such AI-enabled solutions should be carried out to confirm the potential savings in cancer management costs[24].

Our study has several limitations. MSIntuit was developed and validated solely on slides from surgical specimens. With the recent promising results of NICHE-2 trial, neoadjuvant immunotherapy may become the standard of care for CRC patients harbouring MSI phenotype, thus making dMMR/MSI diagnosis on biopsies ubiquitous[25]. Given that MMR-IHC (four immunostainings) and PCR-MSI consumes tissue, using such tools on biopsies would be of particular interest, especially in the case of unresectable CRC where tissue from biopsy specimens can be in very limited supply. While MSIntuit has not been validated on biopsies yet, Echle et al. reported good performance when transferring a model trained to identify MSI on resection specimens on a cohort of biopsies[18]. Although suggesting that MSIntuit could also work on biopsies, further validation on these specimens must be carried out to confirm this hypothesis. At last, MSIntuit calibration requires 30 MSI slides, which can sometimes be difficult to obtain in small centres. Albeit routinely used for many medical devices, calibration might hinder clinical deployment of such tools. Further work needs to be carried out to ensure AI models are agnostic to variability in data acquisition across centres.

With the increasing number of biomarkers to be routinely tested in clinical practice, the need for tools that can both ease and ramp up biomarker testing is paramount. Our tool represents the first step towards the development of AI-based solutions that could identify actionable biomarkers from a single H&E slide used in clinical routine, bringing us closer to reaching the full potential of precision medicine.

## Methods

### Ethical compliance

All experiments were conducted in accordance with the General Data Protection Regulation (GDPR) and the French laws and regulations. Medipath data subjects have generally been informed for the re-use of their samples and data collected during the care for research purposes. Medipath has obtained an approval of the "Ministere de l'Enseignement Superieur, de la Recherche et de l'Innovation (MESRI)" for the storage of samples for research purposes and has nominatively reinformed patients for the re-use of their data for the experiments described in this study. Patients were not compensated for their participation in the study.

### Cohort description

Three cohorts were used in our study: a discovery cohort to train our model, an independent development cohort to gain insights about the model performance, and an independent validation cohort, blinded to patients' MSI statuses, to assess the performance of MSIntuit in a one-shot fashion. Inclusion criteria for all cohorts were as follows: unequivocal histological diagnosis of CRC, available histological slides of resected specimens from the primary tumour, available MSI status. The discovery cohort, denoted TCGA here, is a multicentric cohort of 859 whole slide images (WSI) from 434 patients from the Colon Adenocarcinoma project of TCGA (TCGA-COAD) diagnosed in 24 US centres[26]. 427 Formalin-Fixed Paraffin-Embedded (FFPE) and 432 snap

frozen H&E-stained WSIs from these patients associated with MSI-PCR status were used to develop our model. TCGA slides were digitised at a microns per pixel resolution of 0.25 or 0.5. The Pathology AI Platform (PAIP) cohort was used as a development set and comprised colorectal tumour samples of $n = 47$ patients, collected from three centres in South Korea[27]. PAIP slides were digitised at a MPP resolution of 0.25. The MSI status of these patients was determined using MSI-PCR assays. The validation cohort used for the blind validation consisted of 600 anonymised FFPE H&E WSIs of 600 consecutive resected CRC diagnosed at Medipath pathology laboratories (France) in 2017 and 2018. Patients were originally treated in more than ten centres in France. Tumour samples from these patients were sent to Medipath laboratories. For each patient, one H&E slide was chosen following our guidelines (Supplementary methods). Slide samples were prepared in one single technical platform using the workflow of clinical routine. All slides were digitised at a MPP resolution of 0.25 using two scanners, Philips UFS (Philips, Amsterdam, The Netherlands) and Ventana DP200 (Roche Diagnostics GmbH, Mannheim, Germany), leading to two sets of 600 WSIs referred to as MPATH-UFS and MPATH-DP200. dMMR status was assessed using MMR-IHC for the four MMR proteins, and confirmed by MSI-PCR for n = 33 indeterminate cases (doubt in MMR-IHC interpretation or suspicion of Lynch Syndrome). Clinicopathologic features of these three cohorts can be found in Supplementary Table 10.

### Preprocessing of whole-slide images

A preprocessing pipeline was applied to reduce dimensionality and clean the data before training any model (Fig. 6a). The first step of our pipeline consisted of detecting the tissue on the WSI: a U-Net neural network was used to segment part of the image that contains relevant matter, and discard artefacts such as blur, pen marker etc., as well as the background[28]. U-Net is a fully convolutional neural network architecture widely used for biomedical image segmentation tasks. This U-Net network was previously trained on 460 H&E and IHC slides from an internal dataset where tissue was manually annotated, and validated on 115 slides with a Dice score of 0.96. This network was applied on images of size $2048 \times 2048$ μm ($512 \times 512$ px, at a resolution of 4 MPP) extracted from the WSI. The second step consisted of splitting the slide into smaller images, called "tiles", of $112 \times 112$ μm ($224 \times 224$ px, at a resolution of 0.5 MPP). At least 50% of the tile had to be detected as foreground by the U-Net model to be kept. For training, a maximum of 8,000 tiles were extracted from each slide while all tiles were extracted for inference. The final step consisted of extracting features from each tile: 2048 relevant features were extracted using a wide 50-layer residual net (ResNet50) network (the bottleneck number of channels is twice as large in every block) trained in a self-supervised fashion with Momentum Contrast (MoCo) v2[12,29]. This network was trained on four million tiles from the TCGA-COAD dataset, with heavy data augmentation (random cropping, random flips, colour jitter, random grayscale, gaussian blur), and without using any labels. Feature extractor weights were frozen both for inference and training.

### Performance metrics

The clinical value of the models was evaluated using sensitivity, specificity, and negative predictive value (NPV) metrics. Raw performance of the models was also evaluated using the AUROC. Confidence intervals were generated using bootstrapping with 1000 repetitions.

### Automated quality check

Automated quality check (QC) consisted of two steps: detection of large artefact regions and detection of tumour regions (Figs. 1b, 6b). For the first step, artefact regions such as blurry areas were discarded thanks to the U-Net described in the previous section *Preprocessing of whole-slide images*. The tissue mask generated by the U-Net was then

## a  Preprocessing

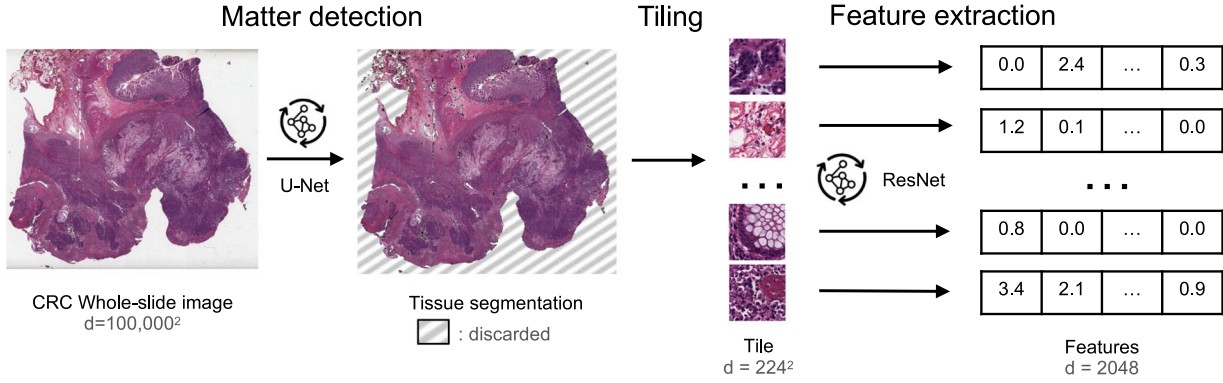

## b  Tumour detection quality check

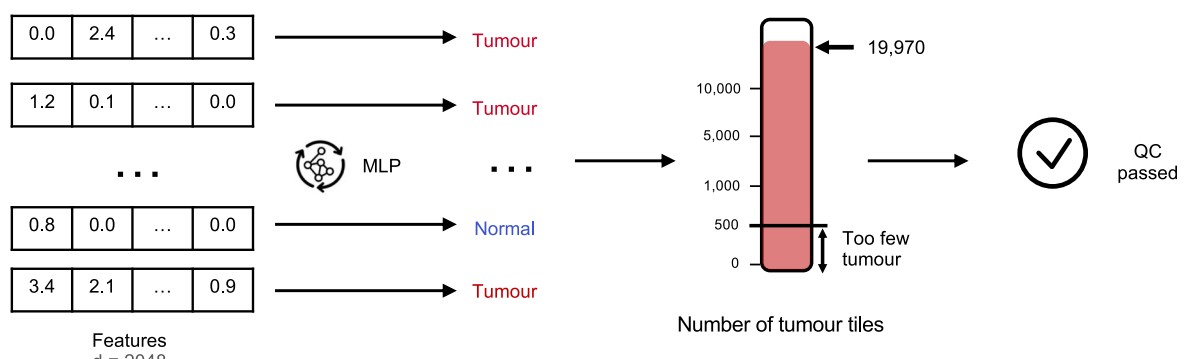

## c  MSI Prediction (Chowder)

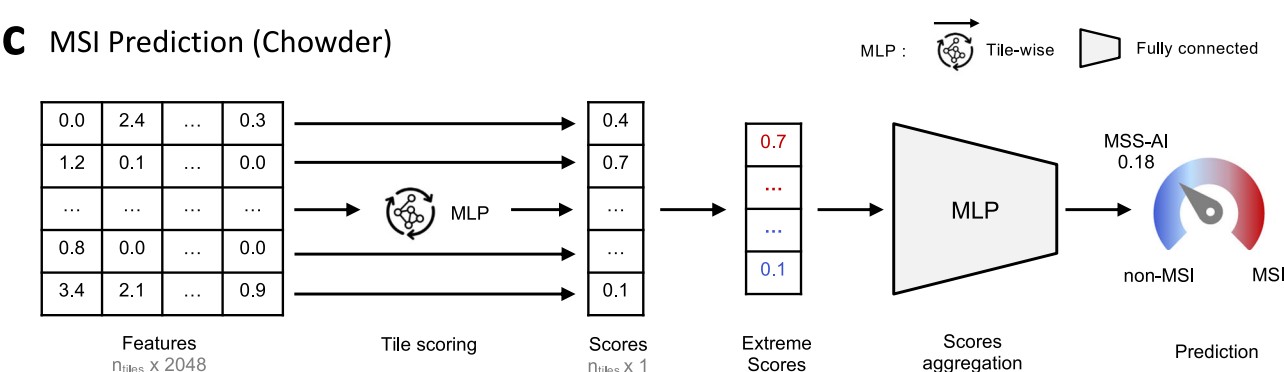

**Fig. 6 | MSIntuit processing pipeline. a** Whole slide image preprocessing: first, a U-Net neural network is used to segment part of the image that contains tissue, and discard the background as well as artefacts. The detected tissue is then split into smaller images called tiles, of 224 × 224 pixels. For each tile, a feature vector of size 2,048 is then extracted using a wide ResNet50 pre-trained with self-supervised learning on 4 million colon cancer images. **b** Tumour detection quality check: second, a MLP previously trained to distinguish tumour from normal tiles is used to identify tumour tiles, using the features generated in step a) as input. If the number of tumour tiles detected is above 500, the quality check is passed, otherwise the slide is discarded. **c** MSI prediction with Chowder: third, the features generated in step a) are used of an MLP that assigns a score to each tile. The ten top and bottom scores are concatenated and used as input for another MLP that aggregates the scores to output a prediction for the slide. Source data are provided as a Source Data file.

briefly examined by a technician to check if there were regions with large artefacts, potentially leading to a new digitisation of the slide. For the second step, a tumour detection model was applied to determine which tiles were tumoural and which tiles were not. This model is a multilayer perceptron (MLP) with one hidden layer of 256 neurons with Rectified Linear Unit (ReLU) activation, trained using the features generated at the end of the preprocessing step (see *Preprocessing of whole-slide images*) of 642,122 tiles from 50 tumour annotated slides of TCGA-COAD. A minimum number of 500 tumour tiles, which corresponds to approximately 6 mm², was set as the cut-off to pass QC,

based on empirical evidence obtained from the development cohorts (Supplementary Fig. 5).

### Model description
A variant of the Chowder model was trained on the discovery cohort to predict MSI status (output) from slide features (input) generated at the end of the preprocessing step (see Preprocessing of whole-slide images, Fig. 6c)[30]. The first layer of Chowder is an MLP with 128 hidden neurons and sigmoid activation that was applied to each tile's features to output one score. The ten top and bottom scores were then

concatenated and fed into an MLP with 128 and 64 hidden neurons and sigmoid activations. The model was trained with binary cross entropy as loss, with weights balanced with respect to the prevalence of MSI in the discovery set.

### Calibration step

To address the issue of variations in data acquisition protocols (e.g. stainers or scanners) that may impact deep learning model prediction distributions, we used a calibration step. This step ensured that MSIntuit yielded a clinically relevant sensitivity without sacrificing the tool's specificity. For both MPATH-DP200 and MPATH-UFS, 30 slides from the same randomly selected dMMR/MSI patients were used to define the operating threshold leading to 1/30 misclassification (meaning, one slide was classified as "MSS-AI", and 29 were classified as "Undetermined"). The impact of the slide selection during the calibration step is reported in Supplementary Table 11. The number of slides used in this calibration step was chosen after a sensitivity analysis on several internal datasets showed that 30 slides were sufficient to ensure with a high likelihood that the sensitivity of MSIntuit on the validation set was between 0.93 and 0.97. In clinical practice, the calibration step is handled by MSIntuit's technical team, and is a prerequisite for any installation in a new pathology laboratory.

### Tool's consistency across slides from different blocks of the same tumour

For a subset of 200 patients of MPATH-DP200 dataset, one to four other tumour slides coming from different blocks of the same surgical resection were digitised, resulting in a total of 398 additional slides. We characterised the tumour morphology of these slides using a ResNet18 model from the TIAToolbox library trained to classify each tile into one of the following categories: adipose tissue, debris, lymphocytes, mucus, smooth muscle, normal colon mucosa, cancer-associated stroma, colorectal adenocarcinoma[31]. We then assessed the variations in MSIntuit predictions according to the slide chosen to be processed by the tool. We also determined how each tissue type category impacted MSIntuit prediction, and which kind of slide was preferable to be selected for MSIntuit processing.

### Statistical analyses

Pearson's correlation coefficient was used to assess the correlation of MSIntuit's scores on the two scanners. Cohen's kappa statistic was used to assess the agreement between MSIntuit's predicted classes across the two scanners and the patients' MSI status. Fleiss' Kappa statistic was used to study the agreement of MSIntuit's predicted classes for the same slides digitised eight times and patient's MSI status. McNemar's test was used to assess the significance of performance difference by selecting in each tumour, the slide with highest and lowest amount of each tissue category. All tests were two-tailed and $p$-values < 0.05 were considered statistically significant.

### Comparison of MSIntuit with MSPath scoring system

We compared the performance of MSIntuit against MSPath on cases from MPATH-DP200 cohort. MSPath is a scoring system that measures the probability of a tumour to be MSI using the following clinicopathological variables: the age at diagnosis, the anatomical site, the histologic type, the grade, the presence of Crohn-like reaction and the presence of tumour infiltrating lymphocytes (TILs)[20]. The age at diagnosis, the anatomical site, the histologic type and the grade were retrieved from clinical data. Cases with one of these variables missing were excluded, leaving 202 cases for the analysis. The assessment of Crohn-like reaction and TILs was divided between three pathologists (D.E. and two experts: K.v.L, M.S) and was conducted following MSPath guidelines. Before starting the assessment on the whole cohort, five cases were scored independently by the three pathologists. The assessment of TILs was discordant for two cases; these cases were

therefore reviewed in a collegial manner to ensure the assessment would be performed in a homogeneous way. The analysis was then performed on the remaining cases by the pathologists blinded to the MSI status, using the same slide as the one processed by MSIntuit.

### Reporting summary

Further information on research design is available in the Nature Portfolio Reporting Summary linked to this article.

## Data availability

All images and the associated MSI status for the TCGA cohort used in this study are publicly available at https://portal.gdc.cancer.gov/ and cBioPortal (https://www.cbioportal.org/). Deidentified pathology images and annotations from the PAIP cohort can be obtained via appropriate data access requests at http://www.wisepaip.org/paip. Datasets MPATH-DP200 and MPATH-UFS are the property of Owkin, France and are available upon request for academic use only. Source data are provided with this paper.

## Code availability

An implementation of the U-Net is available at https://github.com/milesial/Pytorch-UNet. An implementation of MoCov2 is available at https://github.com/facebookresearch/moco. An implementation of Chowder algorithm is made available at https://github.com/CharlieCheckpt/msintuit (https://zenodo.org/badge/latestdoi/670039349)[32].

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

## Acknowledgements

The results published here are in whole or part based upon data generated by the TCGA Research Network: https://www.cancer.gov/tcga.

Regarding the PAIP dataset: De-identified pathology images and annotations used in this research were prepared and provided by the Seoul National University Hospital by a grant of the Korea Health Technology R&D Project through the Korea Health Industry Development Institute (KHIDI), funded by the Ministry of Health & Welfare, Republic of Korea (grant number: HI18C0316). This work was granted access to the HPC resources of IDRIS under the allocation AD011012519 made by GENCI. We thank Pierre Courtiol, Simon Jégou, Benoit Schmauch, and Olivier Moindrot for their contribution in the early development of the model. We thank Sanjana Vasudevan for her corrections of the manuscript. We also thank Dr Alicia Tourneret, Dr Damienne Declerck, Céline Coppolani, Pauline Mespoulhe, and Caroline Rancati for their help in collecting data for the validation cohort.

## Author contributions

Study conception and design: CS, AF, M. Sefta, MA; data collection: TG, AA, SC, JR, DE, SR; Software: CS, RD, OT, NL; analysis and interpretation of results: CS, RD, OT, TG, AA, SC, JR, DE, KvL, M. Svrcek, JNK; draft manuscript preparation: CS, AF, LG, OT, CW. All authors reviewed the results and approved the final version of the manuscript.

## Competing interests

C.S., R.D., O.T., N.L., K.v.L., C.W., M. Sefta, M.A., L.G., A.F. are employees of Owkin Inc. T.G., A.A., S.C., J.R., S.R. are employees of Medipath. J.N.K. declares consulting services for Owkin, France, for Panakeia Technologies, UK, and for DoMore Diagnostics, Norway. J.N.K. declares honoraria for Roche, Eisai, Fresenius. M. Svrcek declares consulting services for Owkin, France. The remaining authors declare no competing interests.
