## [Peer Review File · Nature Communications]

REVIEWER COMMENTS

Reviewer #1 (Remarks to the Author): expertise in bioinformatics image analysis and deep learning

This article presents a study validating the use of the MSIntuit software developed by Owkin on a cohort of 600 patients for the detection of Microsatellite Instability (MSI) in colon cancers from histopathological images.

The study done is very solid, with data obtained from different materials (2 types of scanner). All the analyses are statistically correct and do not present any bias. They highlight the qualities of the tool that the authors wish to show. The method presented appears to be effective and usable in a clinical scope.

However, first of all, some details should be added about the cohort used. As the analysis aims to demonstrate that the tool can be used in a clinical (multicentre) scope, it should be made clear how the pre-sampling and sample preparation were carried out. The text indicates that the samples were all taken from MediPath, so can we still talk about multicentre data?

Secondly, the description of the method should be much clearer. One can guess which operations are performed and with which architectures by digging into the extremely brief description in the article and the supplementary materials, which makes it really difficult to understand. The authors need to add a complete diagram of the process of what is done by the tool: extraction of the tissue by UNet, classification of the tiles by a Chowder+MLP architecture to obtain the cancerous areas, QC check to know if there are enough tumours, then Chowder+MLP again, but trained how? On which data to get the MSI vs. non MSI classifier? We understand how the feature extractor was trained but not the final classifier.

Furthermore, in Figure 1, the authors talk about a threshold "Next, 30 dMMR/MSI WSIs were selected randomly and used to define an appropriate threshold (step 3). Finally, MSIntuit prediction was performed on the remaining slides using the threshold [...]". Who defines this threshold and how? How is it used afterwards? Also, it seems from the figure that it is dependent on the device used to make the scan. If so, what is its importance and influence on the quality of the results? If I want to use this software in my hospital with my own equipment, how should I set this parameter and how can I be sure that it is correct?

Finally, the only comparison is made with "a method which uses a feature extractor 93 trained on ImageNet dataset in a supervised fashion (ImageNet)". It is not known which method was trained, nor how. Moreover, comparing a network trained on natural images (ImageNet) to another one specialised on histopathological images of colon cancer is not very fair. Would we have similar results with a classical supervised approach trained on colon cancer images?

In conclusion, this is a paper that presents a series of experiments that show that the MSIntuit software can be used for the MSI vs. non-MSI classification task on (pseudo)-multicentric data. However, the experiments are not at all reproducible as the description of the whole method is not clearly presented and the code is not provided. This is particularly annoying since the authors are themselves members of the company that develops the software. To be a more interesting contribution to the scientific world, it is necessary to better present the method used and to better compare it to other current state-of-the-art approaches in the field.

Reviewer #2 (Remarks to the Author): expertise in colorectal cancer biomarkers

Manuscript Review

Title: "Blind validation of MSIntuit, an AI-based pre-screening tool for MSI detection from histology

slides of colorectal cancer
Reviewers: Michael Hall, Dina Ioffe

SUMMARY

In this manuscript, the authors report results of validation studies conducted on an AI-based pathology imaging tool called MSIntuit which is designed to detect MSI from stained pathology slides (H&E). MSIntuit was developed with a discovery cohort (TCGA), an independent development cohort, and an independent validation cohort. All cohorts were CRC and known to be MSI-H by PCR. A Korean cohort called PAIP was used for development, and all samples were MSI-H by PCR. The validation cohort was H&E slides from 600 consecutive resected CRCs. Slides were digitized with two scanners. They describe various parts of the process and the performance of the different components (e.g. the scanner types, the automated quality check, calibration check). They also look at how selection of different slides with varying degree of different cell types (e.g. mucous, smooth muscle, etc) affected the tool performance. On the DP200 scanner, sensitivity was 98% (95-100%) for MSI-H and specificity 46% (42-50%). On the UFS scanner sensitivity was 96% (91-98%) with a specificity of 47% (43-51%). The authors also found strong reproducibility and consistency of MSI-H calls when compared between scanners. The authors found by examining different tissues that particular features (inflammation and mucin and poor differentiation) were more often MSI, but normal well-differentiated glands were not—however they found mucin could also be a false positive and that more tumor had higher specificity. High concordance across scanners and between blocks is important for generalizability of findings and demonstrating that MSIntuit has potential for implementation.

GENERAL

The manuscript is generally well-written and written concisely, which is a plus. Very interesting and promising data that has the potential to make a significant impact in clinical practice and optimizing diagnostics. Nice images.

Applied AI/machine learning in medicine is a hot topic and is relevant to the Nature Communications readership.

Lynch syndrome is common and remains under detected--developing new tools to identify patients is critical. The goal of preserving scarce tumor tissue using new tools is also very relevant to the practice of oncology.

The dissemination potential of this tool is very high—no molecular lab needed—so this increases the generalizability of these findings considerably.

Use of multiple cohorts in development and testing of tool is a strength.

Ample Tables and Figures to support research.

Biggest strength(s): Very interesting work that is timely and relevant to clinical practice, is generalizable given its inter-scanner and intra-tumoral reliability, as well as validation using a large sample size.

Biggest weakness(es): The paper is somewhat confusing with the Methods section following the Results section, especially since some abbreviations/terms are not defined in the results section. Having the information from the Methods section informs the Results section and makes the paper much easier to read and understand.

SPECIFIC COMMENTS

MAJOR

1-The manuscript is generally well written, but there are unclear statements without explanation or reference.

- Line 84 "this metric (AUROC) can hide a severe lack of generalization and is not relevant to clinical practice"—not clear what this means?
- Line 91 "sensitivity is maintained at new sites and on new cohorts" –what does sites refer to?

Disease sites? Geographic locations?

2-At several points throughout the manuscript there are references to “model generalizability” but the authors do not explain what this means in clinical terms?

3-It’s unclear to me whether the “automated quality check” described is built into the MSIntuit tool? Or does this QC occur before the tool is used? Or is this a function linked to the individual scanners?

Similarly, the check for % tumor? Was this function also built into the tool or is this somehow separate?

If this QC check is of benefit, please clarify how not doing this would affect results (e.g. if blurry regions were not excluded or slides with too little tumor were included?). Does the need for this quality check, which weeds out lower quality images and slides, decrease the generalizability of this tool beyond the study reported here (e.g. if blurry images/slides on scanners and small samples are more common in the real world than they were in this study)?

3-Many of the features that the tool identifies as being suggestive of MSI-H are ones that are already known and recognizable by a skilled pathologist? It would be valuable to see how the tool performs compared to a senior pathologist?

4-Sensitivity analyses are not reported when the degree of MSI (e.g. 5/5 vs 2/5 microsatellites unstable). MSI in CRC tends to be very robust compared to other tumors—but this is not true for other tissues, where MSI may be more subtle. It would be important to see if the tool is sensitive for low level MSI, or tumors with MSH6 mutations may be more often missed.

5-Local calibration is said to be a necessary step for any site that would use MSIntuit? It’s not clear to me why this would have to be done at each site?

6-The authors make the point that correlations of predictions between the two scanners was stronger than the correlation using an ImageNet pre-trained feature extractor. Can the authors expand upon why MSIntuit extracts features better than this other modality? Can the authors explain in more depth what “self-supervised learning” is and why this was more successful than ImageNet-based training?

7-In the Discussion, consider emphasizing the decreased time to MSI results with MSIntuit, as well as the potential to preserve limited tissue for other testing (if validated in biopsies), as this is clinically relevant.

8-Consider adding a hypothesis regarding why including whole slide rather than tumor only improved AI performance since this is counter intuitive.

MINOR (by line #)

Abstract – specificity range is 47-46%, is that a typo?

47: Digitizing (not digitising)

60: Replace “frequent” with “common”

112: Allows [us] to detect slides

Additional: Clarify abbreviations and terms that may not be well known to all clinical readers (e.g., TCGA in abstract, PAIP & UNet in methods, feature extractor in results, CE-IVD in discussion).

RESPONSE TO REVIEWERS' COMMENTS

We would like to thank the reviewers for their insightful comments that helped us improve our manuscript. Please find attached a point-by-point response to their comments and the modified version of the manuscript. Any modifications are highlighted in red.

Reviewer #1 (Remarks to the Author): expertise in bioinformatics image analysis and deep learning

This article presents a study validating the use of the MSIntuit software developed by Owkin on a cohort of 600 patients for the detection of Microsatellite Instability (MSI) in colon cancers from histopathological images. The study done is very solid, with data obtained from different materials (2 types of scanner). All the analyses are statistically correct and do not present any bias. They highlight the qualities of the tool that the authors wish to show. The method presented appears to be effective and usable in a clinical scope.

Thank you very much for highlighting the relevance and the scientific rigor of our study. We have addressed every single point that you mentioned, see below.

However, first of all, some details should be added about the cohort used. As the analysis aims to demonstrate that the tool can be used in a clinical (multicentre) scope, it should be made clear how the pre-sampling and sample preparation were carried out. The text indicates that the samples were all taken from MediPath, so can we still talk about multicentre data ?

Yes, the tumor samples in the validation cohort were obtained from multiple centers in the Medipath network between 2017 and 2018. Medipath is one of the largest pathology networks in France with 30 pathology labs across the country. The respective patients were originally treated in more than ten centers in France, meaning that the cohort covers a relatively wide spectrum of the French population. Tumor samples from these cases were then sent to a central Medipath laboratory, were processed and scanned with two scanners to compare the impact of scanner hardware. We have clarified this in the revised manuscript as follows (line 284): "Patients were originally treated in more than ten centres in France. Tumour samples from these patients were sent to Medipath laboratories. [...] Slide samples were prepared in one single technical platform using the workflow of clinical routine."

Secondly, the description of the method should be much clearer. One can guess which operations are performed and with which architectures by digging into the extremely brief description in the article and the supplementary materials, which makes it really difficult to understand. The authors need to add a complete diagram of the process of what is done by the tool: extraction of the tissue by UNet, classification of the tiles by a Chowder+MLP architecture to obtain the cancerous areas, QC check to know if there are enough tumours, then Chowder+MLP again, but trained how? On which data to get the MSI vs. non MSI classifier? We understand how the feature extractor was trained but not the final classifier.

We agree that a clear and exhaustive description of the methods is paramount for reproducibility. We have made a substantial effort to clarify all of these points in the revised manuscript. Specifically,...

- We added a figure in the main text illustrating the overall pipeline (new figure 6).
- We also moved the paragraph "*Preprocessing of whole-slide images*" from the suppl. Methods to the main methods section (line 295)
- We added more detail in the paragraphs "*Automated Quality Check*" and "*Model description*".

- Regarding the training of MSI vs. non-MSI classifier, our method was based on our previous work on the Chowder pipeline. The revised manuscript now reads (line 333) : “A variant of Chowder model was trained on the discovery cohort to predict MSI status (output) from slide features (input) generated at the end of the preprocessing step [...]”.

Altogether, we are confident that these modifications will help the readers better understand the different steps involved in our pipeline, making our approach reproducible and trustable.

Furthermore, in Figure 1, the authors talk about a threshold "Next, 30 dMMR/MSI WSIs were selected randomly and used to define an appropriate threshold (step 3). Finally, MSIntuit prediction was performed on the remaining slides using the threshold [...]". Who defines this threshold and how? How is it used afterwards? Also, it seems from the figure that it is dependent on the device used to make the scan. If so, what is its importance and influence on the quality of the results? If I want to use this software in my hospital with my own equipment, how should I set this parameter and how can I be sure that it is correct?

We thank the reviewers for highlighting this critical point. Indeed, any deep learning system in pathology requires a threshold to convert continuous prediction values into actionable categories. There is no universally agreed upon method to determine such thresholds. The optimal threshold is known to vary between laboratories (Kleppe et al., ESMO Open 2022) due to batch effects (Howard et al., Nat Comms 2021). Some authors (Echle et al., ESMO Open, 2022) have simply pre-defined an arbitrary threshold value of 0.5. However, this results in a suboptimal performance. Other authors evade defining a threshold and just report the AUROC, which is however not clinically usable, as we highlighted in the introduction.

In contrast, MSIntuit uses a calibration step to determine the optimal threshold for each laboratory. We obtain 30 dMMR/MSI WSI from each site to find a threshold that guarantees a clinically relevant sensitivity. Specifically, once the 30 WSIs are obtained, the MSIntuit continuous score is computed for each WSI, and the 2nd smallest score is used as the threshold, which results on average in a sensitivity of 93-97%. MSIntuit is then configured to use this threshold in future inferences to output the predicted status: any new WSI with a MSIntuit score below this threshold will be labeled as ‘MSS-AI’. This unique procedure makes our assay safe. This calibration step is handled by MSIntuit’s technical team and is a prerequisite for any new installation. This procedure is part of the specifications of MSIntuit that are laid down in the regulatory documents for CE/IVD approval so it is a core component of our methodology.

We have rephrased parts of the methods section to highlight this accordingly.

Finally, the only comparison is made with "a method which uses a feature extractor trained on ImageNet dataset in a supervised fashion (ImageNet)". It is not known which method was trained, nor how. Moreover, comparing a network trained on natural images (ImageNet) to another one specialized on histopathological images of colon cancer is not very fair. Would we have similar results with a classical supervised approach trained on colon cancer images?

We thank the reviewer for their feedback. The comparison analysis consisted of changing MSIntuit feature extractor by one pre-trained on ImageNet, keeping all other steps of the MSIntuit pipeline unchanged. We clarified it in the results section.

We agree that comparing our feature extractor pre-trained with self-supervised learning on histology images against one pre-trained on ImageNet, which does not contain any histology image, may seem unfair and the relevance of this comparison was not explained enough in the manuscript. This comparison was performed because there are no annotated datasets of histology images equivalent to ImageNet in terms of size (1.2 million images) and annotation

diversity (1,000 classes). That is why a large number of computational pathology studies have used a feature extractor pre-trained on ImageNet over the past few years (Coudray et al., Nat Med, 2018; Courtiol et al., Nat Med, 2019; Shmatko et al., Nat Cancer 2022). Therefore, we believe that this comparison with an algorithm pre-trained on ImageNet is useful to benchmark ourselves against what has been done in prior studies.

Nevertheless, the alternative remains to pre-train models on labeled histology datasets, even if those do not reach the breadth of ImageNet. We have addressed this point by performing additional experiments which we included in the revised manuscript. Specifically, we used a “wide ResNet-50” from the TIAToolbox software library that was pre-trained on NCT-CRC-100K, a dataset of 100,000 colorectal cancer images where the prediction task is to identify nine tissue classes (Pocock et al., Comms Med, 2022; Kather et al., Plos Medicine 2019). This model was used as a feature extractor and compared to our “wide ResNet-50” model which was pre-trained with MoCo v2 on colon cancer images, all other steps being the same (training dataset, matter detection, quality check, downstream prediction). Our experimental results show that the MSIntuit procedure outperforms all other tested approaches. The following table has been added to the revised manuscript (Supplementary table 1), and the results section has been updated to reflect this.

Pre-training dataset	Method	Block	TCGA	PAIP	MPATH-DP200	MPATH-UFS
ImageNet	Supervised	Penultimate	0.80 +- 0.05	0.92 [0.84-0.97]	0.79 [0.74-0.83]	0.78 [0.73-0.83]
		Last	0.81 +- 0.04	0.88 [0.73-0.98]	0.78 [0.73-0.82]	0.73 [0.67-0.77]
NCT-CRC-100K	Supervised	Penultimate	0.79 +- 0.06	0.81 [0.67-0.92]	0.79 [0.75-0.83]	0.68 [0.62-0.73]
		Last	0.77 +- 0.04	0.72 [0.56-0.86]	0.71 [0.66-0.76]	0.61 [0.56-0.67]
TCGA-COAD	Self-supervised (MSIntuit)	Last	0.93 +- 0.03	0.97 [0.90-0.99]	0.88 [0.84-0.91]	0.87 [0.83-0.90]

In addition, we now compare our pipeline to the previously published “iDaRS” method by the TIA lab at Warwick university (Bilal et al., Lancet Digital Health, 2021). As can be seen in the table below, our model outperformed it on the three external cohorts (PAIP: Delong p=0.06, MPATH-DP200: p=0.001, and MPATH-UFS: p<0.001). We have added this information to the revised manuscript as well (Supplementary table 2).

	PAIP	MPATH-DP200	MPATH-UFS
iDARS (TIAToolbox)	0.86 [0.75-0.94]	0.80 [0.76-0.85]	0.76 [0.71-0.81]
MSIntuit	0.97 [0.90-0.99]	0.88 [0.84-0.91]	0.87 [0.83-0.90]

In conclusion, this is a paper that presents a series of experiments that show that the MSIntuit software can be used for the MSI vs. non-MSI classification task on (pseudo)-multicentric data. However, the experiments are not at all reproducible as the description of the whole method is

not clearly presented and the code is not provided. This is particularly annoying since the authors are themselves members of the company that develops the software. To be a more interesting contribution to the scientific world, it is necessary to better present the method used and to better compare it to other current state-of-the-art approaches in the field.

Thank you very much for highlighting the robustness of our experimental approach. We share an implementation of the Chowder model in the revised version of the manuscript : “An implementation of Chowder is available at <https://github.com/CharlieCheckpt/msintuit>” (line 482).

We hope that our additional experiments have satisfied your requests. We would like to stress that unlike all other previously published methods on Deep Learning based MSI detection (Echle et al., *Gastroenterology*, 2020; Yamashita et al., *Lancet Oncology*, 2021; Bilal et al., *Lancet Digital Health* 2021, and many others), our approach has received regulatory approval and can be used according to the product specifications for routine diagnostic use in the European Union.

Reviewer #2 (Remarks to the Author): expertise in colorectal cancer biomarkers

Manuscript Review

Title: “Blind validation of MSIntuit, an AI-based pre-screening tool for MSI detection from histology slides of colorectal cancer

Reviewers: Michael Hall, Dina Ioffe

SUMMARY

In this manuscript, the authors report results of validation studies conducted on an AI-based pathology imaging tool called MSIntuit which is designed to detect MSI from stained pathology slides (H&E). MSIntuit was developed with a discovery cohort (TCGA), an independent development cohort, and an independent validation cohort. All cohorts were CRC and known to be MSI-H by PCR. A Korean cohort called PAIP was used for development, and all samples were MSI-H by PCR. The validation cohort was H&E slides from 600 consecutive resected CRCs. Slides were digitized with two scanners. They describe various parts of the process and the performance of the different components (e.g. the scanner types, the automated quality check, calibration check). They also look at how selection of different slides with varying degrees of different cell types (e.g. mucous, smooth muscle, etc) affected the tool performance. On the DP200 scanner, sensitivity was 98% (95-100%) for MSI-H and specificity 46% (42-50%). On the UFS scanner sensitivity was 96% (91-98%) with a specificity of 47% (43-51%). The authors also found strong reproducibility and consistency of MSI-H calls when compared between scanners. The authors found by examining different tissues that particular features (inflammation and mucin and poor differentiation) were more often MSI, but normal well-differentiated glands were not—however they found mucin could also be a false positive and that more tumor had higher specificity. High concordance across scanners and between blocks is important for generalizability of findings and demonstrating that MSIntuit has potential for implementation.

We would like to thank the reviewer for pointing out the merit of our study. We have addressed all remaining points in the revised manuscript, see below.

GENERAL

The manuscript is generally well-written and written concisely, which is a plus. Very interesting and promising data that has the potential to make a significant impact in clinical practice and optimizing diagnostics. Nice images.

Thank you. No action needed.

Applied AI/machine learning in medicine is a hot topic and is relevant to the Nature Communications readership.

Thank you. No action needed.

Lynch syndrome is common and remains under detected--developing new tools to identify patients is critical. The goal of preserving scarce tumor tissue using new tools is also very relevant to the practice of oncology.

We agree. Screening for Lynch syndrome is a hugely relevant area of oncology which is not optimally implemented in clinical workflows today.

The dissemination potential of this tool is very high—no molecular lab needed—so this increases the generalizability of these findings considerably.

Thank you. No action needed.

Use of multiple cohorts in development and testing of tool is a strength.

Thank you. No action needed.

Ample Tables and Figures to support research.

Thank you. No action needed.

Biggest strength(s): Very interesting work that is timely and relevant to clinical practice, is generalizable given its inter-scanner and intra-tumoral reliability, as well as validation using a large sample size.

Thank you very much.

Biggest weakness(es): The paper is somewhat confusing with the Methods section following the Results section, especially since some abbreviations/terms are not defined in the results section. Having the information from the Methods section informs the Results section and makes the paper much easier to read and understand.

We agree that the article structure where the Methods section appears first is more common but the Nature Communications format imposes that the methods are at the end of the manuscript.

SPECIFIC COMMENTS

MAJOR

1-The manuscript is generally well written, but there are unclear statements without explanation or reference.

Thank you very much. We have clarified all these points by revising the manuscript carefully.

- Line 84 “this metric (AUROC) can hide a severe lack of generalization and is not relevant to clinical practice”—not clear what this means?

We refer to Kleppe et al. (ESMO Open, 2022) who pointed out that the AUROC can be high even for a model that generalizes poorly. Therefore, to make sure that our MSIntuit approach yields a robust performance and generalizes well, we have employed a unique calibration strategy. We have added this information to the revised manuscript, added the above-mentioned citation and explained the importance of model generalization and of considering other metrics besides the AUROC (lines 90-99).

• Line 91 “sensitivity is maintained at new sites and on new cohorts” –what does sites refer to? Disease sites? Geographic locations?

Site referred to geographic locations (pathology laboratories). We have clarified this (line 105): “We therefore propose a method that guarantees the sensitivity is maintained at new pathology laboratories”.

2-At several points throughout the manuscript there are references to “model generalizability” but the authors do not explain what this means in clinical terms?

We added common definition of the term “model generalisability” to the revised manuscript (line 90): “Here, we refer to model generalization as the ability of the model to yield consistent sensitivity and specificity in different independent validation cohorts (e.g. with different ethnicities), under different clinical settings (e.g. with different scanners).”

3-It's unclear to me whether the “automated quality check” described is built into the MSIntuit tool? Or does this QC occur before the tool is used? Or is this a function linked to the individual scanners?

The QC step is part of the tool as mentioned at the end of the introduction line 118 : “Our tool includes an automatic slide quality check [...]”. The figure 6 we added in the revision also illustrates how the QC is used in the MSIntuit pipeline.

Similarly, the check for % tumor? Was this function also built into the tool or is this somehow separate?

The check for the amount of tumor is also built in the tool and is part of the quality check step (see new figure 6b).

If this QC check is of benefit, please clarify how not doing this would affect results (e.g. if blurry regions were not excluded or slides with too little tumor were included?). Does the need for this quality check, which weeds out lower quality images and slides, decrease the generalizability of this tool beyond the study reported here (e.g. if blurry images/slides on scanners and small samples are more common in the real world than they were in this study)?

We thank the reviewer for these suggestions. To clarify this question, we have performed extensive additional experiments. Below, we present an ablation study that we conducted on the MPATH-DP200 cohort of the two QC steps (tumor check and blurry check).

Ablation of tumor check: we kept slides with too few tumour instead of discarding them. This means that 28 slides with small tumor area were added to the validation cohort. As can be seen in the table below, without discarding the slides with too few tumor, performance decreased to an AUROC of 0.86, a sensitivity of 0.96, a specificity of 0.45 and a NPV of 0.98.

Ablation of blurry check: We also kept the slides with large blurry areas (n=13), instead of using the rescanned version. This means that we assessed the model performance on the same

number of slides, but we swapped the rescanned slides for the blurry ones. Although Echle et al., Gastroenterology 2020 reported that blurry slides can lead to misclassifications; in our experiments we did not see a degradation of performance by using the blurry slides (see Table below, which was added to the revised manuscript as new suppl. Table 6). This indicates the robustness of our MSIntuit model with respect to non-optimal images.

	n	AUC	Sensitivity	Specificity	NPV
QC (tumour and blurry check, baseline)	537	0.88 [0.84-0.91]	0.98 [0.95-1.0]	0.46 [0.42-0.50]	0.99 [0.98-1.0]
No tumour check	565	0.86 [0.82-0.89]	0.96 [0.91-0.99]	0.45 [0.42-0.49]	0.98 [0.97-1.0]
No blurry check	537	0.88 [0.85-0.91]	0.98 [0.95-1.0]	0.46 [0.42-0.50]	0.99 [0.98-1.0]

Nevertheless, even though the blurriness check did not improve AUC, Sensitivity, Specificity and NPV (as per the Table above), we observed that blurriness check results in an improved distribution of the prediction scores of individual slides, which were less scattered and therefore more consistent, as shown in the Figure below (new suppl. Figure 2). Specifically, the median predictions for non-MSI cases with blurry (respectively rescanned) slides was 0.29 (respectively 0.21). Median predictions for MSI cases with blurry slides (respectively rescanned) slides were 0.55 (respectively 0.56).

Together, these data show that our QC procedure makes the MSIntuit method more robust and results in a more consistent performance. We have added this information in the revised results section of the manuscript.

Finally, the reviewer asked about the specimen size in the real world. We believe that the validation set presented in this study represents a real world cohort that was not cherry-picked. The slides were collected from consecutive colorectal cancer cases diagnosed during clinical routine at nine pathology labs in 2017 and 2018, minimizing the potential for bias. Furthermore, the pathology lab performed the digitisation of slides using two scanners widely adopted across pathology labs, following the same procedures used in clinical routine. Consequently, we expect that a similar amount of slides will pass QC (95-98%) when using MSIntuit in clinical routine.

3- Many of the features that the tool identifies as being suggestive of MSI-H are ones that are already known and recognizable by a skilled pathologist? It would be valuable to see how the tool performs compared to a senior pathologist?

As the reviewer points out, most of the features that MSIntuit detects are already known to be associated with MSI status. We believe that this is a strength of our method: Without being explicitly told to do so, our trained model has “rediscovered” morphological features which are strongly linked to MSI status.

Indeed, in theory, a pathologist could quantify these features on a checklist and then make a prediction about the MSI status. As Yamashita et al. (Lancet Oncology 2021) have shown, this results in a rather poor performance. To address the reviewer’s concern, we have performed a systematic reader study with three pathologists, including subspecialty experts. In this experiment, we used MSPath, a scoring system that measure the probability of a tumor to be MSI, given several clinical and pathological features: age at diagnosis, the anatomical site, the histologic type, the grade, the presence of Crohn’s-like reaction and the presence of tumor infiltrating lymphocytes (Jenkins et al., Gastroenterology, 2007). We have added the results of this experiment to the revised manuscript (results: line 255, discussion: line 406, methods: line 371). Interestingly, the results of this experiment show that the AUROC of MSIntuit is higher than the AUROC this clinico-pathological scoring system, while the sensitivity/specificity at a pre-defined threshold value is closer, and that the combination of two scores yields better results (line 264).

Nevertheless, we believe that this experiment is more of academic than of practical relevance. The guess of a pathologist, even a senior one, is not an objective or reproducible scoring. Also, it is time-consuming and pathologists would have to be specifically trained to perform it. Given that there is a major shortage of pathologists around the world, we strongly believe that automatic prediction of MSI status with MSIntuit is a much more feasible, scalable and reproducible way of MSI screening than manual assessment by pathology experts.

4- Sensitivity analyses are not reported when the degree of MSI (e.g. 5/5 vs 2/5 microsatellites unstable). MSI in CRC tends to be very robust compared to other tumors—but this is not true for other tissues, where MSI may be more subtle. It would be important to see if the tool is sensitive for low level MSI, or tumors with MSH6 mutations may be more often missed.

We thank the reviewer for the interesting suggestions. We performed extensive additional experiments to investigate these questions.

Specifically, we analyzed the model predictions in populations with different levels of MSI of the discovery cohort (TCGA), which is the only cohort where the PCR status MSS / MSI-L / MSI-H was available for all patients. Interestingly, predictions in the MSI-low population were close to the ones of the MSS population, but still significantly different (t-test $p=0.005$, see plot below). This suggests that the MSI-low population may harbor a different morphology than the MSS population, which is in line with previous studies (Kather et al., Nat Med, 2019).

Furthermore, we also assessed the ability of MSIntuit to detect isolated losses of PMS2 and MSH6 mutations on the validation cohorts. The article now reads line 177: “We assessed the ability of MSIntuit to detect unusual isolated losses of PMS2 and MSH6 mutations, that were found to cause discordance between MMR-IHC and PCR-MSI (De Salins et al., ESMO Open, 2021). MSIntuit reached a sensitivity of 0.91 (respectively 0.91) and 0.67 (respectively 0.72) on MPATH-DP200 (respectively MPATH-UFS) to detect isolated PMS2 and MSH6 losses (Supplementary Table 5). Because less than ten cases displayed these mutation patterns on the two cohorts, further assessment with larger sample sizes should be carried out to confirm these numbers.”

5-Local calibration is said to be a necessary step for any site that would use MSIntuit? It's not clear to me why this would have to be done at each site?

We thank the reviewer for her question regarding the calibration step. Indeed, the threshold computation is an important part of the software installation process in a new center before using MSIntuit in clinical routine. It is well known that Deep Learning-based predictions have batch effects that are dependent on a multitude of factors at different sites (Kleppe, ESMO Open 2022; Howard et al., Nat Comms, 2022). We designed MSIntuit to provide safe and reliable predictions irrespective of such domain shifts. Our approach is to use a rigorous local calibration procedure, by which the decision threshold is specifically adapted to each site. This is a core part of MSIntuit, as we explain in the results section (line 137): “To address the issue of variations in data acquisition protocols that may impact deep learning model prediction distributions, we used a calibration strategy to ensure a sensitivity between 0.93 and 0.97 was obtained for the blind validation”.

6-The authors make the point that correlations of predictions between the two scanners was stronger than the correlation using an ImageNet pre-trained feature extractor. Can the authors expand upon why MSIntuit extracts features better than this other modality? Can the authors explain in more depth what “self-supervised learning” is and why this was more successful than ImageNet-based training?

We thank the reviewer for their comment which can help to clarify the concept of self-supervised learning (SSL) in our manuscript, and its influence on MSIntuit. We have added a brief explanation of SSL in the introduction, line 106: “Self-supervised learning (SSL) has emerged in the computer vision field as a powerful method to learn rich vector representations from images. SSL consists of training a feature extractor to solve a “pretext task”, that is a task that does not require human annotations, as opposed to traditional supervised learning. Such tasks can be reconstructing a part of the image which is masked, or producing similar representations for two augmented versions of the same image (He et al., Arxiv, 2021; Chen et al., Arxiv, 2020). MSIntuit leverages a feature extractor tailored for histology, trained on four million colorectal cancer pathology images with SSL.”.

We also commented on the better scanner robustness of our method, line 208: “As MSIntuit feature extractor was trained specifically to produce similar representations under heavy data augmentations, we believe that this could explain the enhanced robustness of MSIntuit to scanner variations.”

7-In the Discussion, consider emphasizing the decreased time to MSI results with MSIntuit, as well as the potential to preserve limited tissue for other testing (if validated in biopsies), as this is clinically relevant.

We thank the reviewer for the suggestion. We have added this point to the revised discussion accordingly (line 426): “With MMR-IHC turnaround time varying between two to seven days in different clinical settings and MSI-PCR results delivery that can take more than a week, pre-screening for non-MSI patients with an AI-enabled solution in a few hours holds a real potential of time-savings, both for pathologists and patients”, and line 439, “Given that MMR-IHC (four immunostainings) and PCR-MSI consumes tissue, using such tools on biopsies would be of particular interest, especially in the case of unresectable CRC where tissue from biopsy specimens can be in very limited supply.”

8-Consider adding a hypothesis regarding why including whole slide rather than tumor only improved AI performance since this is counter intuitive.

It is known (and has been discussed previously by Echle et al., ESMO Open 2022 and others), that extra-tumoral features are associated with MSI status, such as peritumoral inflammation. We have added this explicit hypothesis in the results section. The article now reads line 249: “Regions predictive of MSI also included inflammation outside the tumour area (25%), which may explain why better performance was obtained better considering the whole-slide and not just the tumour content.”

MINOR (by line #)

Abstract – specificity range is 47-46%, is that a typo?

The specificities were written to match the order of the sensitivity mentioned before, meaning we obtained a sensitivity/specificity of 96% / 47% on one dataset and 97% / 46% on the other dataset. If the reviewer prefers, we can write the metrics in the ascending order instead.

47: Digitizing (not digitising)

Our manuscript was written in UK English, which we believe is allowed by Nature Communications. This is why we wrote “*digitising*” instead of “*digitizing*”.

60: Replace “frequent” with “common”

The article now reads line 64, “[...] the most common form of hereditary predisposition to develop CRC”.

112: Allows [us] to detect slides

The article now reads line 126: “This step allows to automatically detect slides that were not properly scanned [...]”.

Additional: Clarify abbreviations and terms that may not be well known to all clinical readers (e.g., TCGA in abstract, PAIP & UNet in methods, feature extractor in results, CE-IVD in discussion).

We clarified the abbreviations.

The article now reads:

- Line 49: “[...] the first clinically approved artificial intelligence (AI) based [...]”
- Line 50: “[...] from haematoxylin-eosin (H&E) stained slides.”
- Line 50: “After training on samples from The Cancer Genome Atlas (TCGA) [...]”
- Line 67: “[...] approved by the U.S. Food and Drug Administration (FDA)”
- Line 69: “[...] such as the National Institute for Health and Care Excellence (NICE) and the National Comprehensive Cancer Network (NCCN)”
- Line 276: “[...] the Colon Adenocarcinoma project of TCGA (TCGA-COAD)”
- Line 309: “[...] using a wide 50-layer residual net (ResNet50)”
- Line 310: “[...] with Momentum Contrast (MoCo) v2.”
- Line 327: “[...] with Rectified Linear Unit (ReLU) activation”
- Line 280: “[...] on PAIP (Pathology AI Platform)”
- Line 413: “With recent In-vitro Diagnostic ‘Conformité Européenne’ (CE-IVD) certification [...]”

We believe U-Net is not an acronym, but the name of a neural network with a U shape originally proposed by Ronneberger et al., MICCAI, 2015. Chowder is also not an acronym. Unless the reviewer believes some kind of clarification should still be made for these model names, we would prefer leaving these two terms as is.

REVIEWERS' COMMENTS

Reviewer #1 (Remarks to the Author):

The description of the methodology is now much clearer and the authors resolved all the critical points I highlighted. The paper could be published in its new form.

Reviewer #2 (Remarks to the Author):

The authors have been extremely responsive to the comments of both reviewers and have performed additional analyses that have clarified the performance of the MSIntuit tool. They have also added substantial supporting/explanatory text to the manuscript to clarify different elements of the evaluation and output of the MSIintuit tool.

RESPONSE TO REVIEWERS' COMMENTS

Reviewer #1 (Remarks to the Author):

The description of the methodology is now much clearer and the authors resolved all the critical points I highlighted. The paper could be published in its new form.

Thank you, no actions needed.

Reviewer #2 (Remarks to the Author):

The authors have been extremely responsive to the comments of both reviewers and have performed additional analyses that have clarified the performance of the MSIntuit tool. They have also added substantial supporting/explanatory text to the manuscript to clarify different elements of the evaluation and output of the MSIntuit tool.

Thank you, no actions needed.